# Study on the Performance Test of Fe–Ce–Al/MMT Catalysts with Different Fe/Ce Molar Ratios for Coking Wastewater Treatment

**DOI:** 10.3390/molecules29091948

**Published:** 2024-04-24

**Authors:** Xiaoping Su, Xiangtong Wang, Ning Li, Longjian Li, Yilare Tuerhong, Yongchong Yu, Zhichao Wang, Tao Shen, Qiong Su, Ping Zhang

**Affiliations:** Key Laboratory of Environment-Friendly Composite Materials of the State Ethnic Affairs Commission, Gansu Provincial Biomass Function Composites Engineering Research Center, Key Laboratory for Utility of Environment-Friendly Composite Materials and Biomass in University of Gansu Province, Gansu Province Research Center for Basic Sciences of Surface and Interface Chemistry, College of Chemical Engineering, Northwest Minzu University, Lanzhou 730124, China; 13087526096@163.com (X.S.); 17862353822@163.com (X.W.); wxt15054093018@126.com (N.L.); 15253036264@163.com (L.L.); nuanguang-1228@163.com (Y.T.); 18722969475@163.com (Y.Y.); eternal163@outlook.com (Z.W.); kenan98@163.com (T.S.); hgsq@xbmu.edu.cn (Q.S.)

**Keywords:** Fe–Ce–Al/MMT catalyst, catalytic wet peroxide oxidation, COD, coking wastewater

## Abstract

It is very important to choose a suitable method and catalyst to treat coking wastewater. In this study, Fe–Ce–Al/MMT catalysts with different Fe/Ce molar ratios were prepared, characterized by XRD, SEM, and N_2_ adsorption/desorption, and treated with coking wastewater. The results showed that the optimal Fe–Ce–Al/MMT catalyst with a molar ratio of Fe/Ce of 7/3 has larger interlayer spacing, specific surface area, and pore volume. Based on the composition analysis of real coking wastewater and the study of phenol simulated wastewater, the response surface test of the best catalyst for real coking wastewater was carried out, and the results are as follows: initial pH 3.46, H_2_O_2_ dosage 19.02 mL/L, Fe^2+^ dosage 5475.39 mL/L, reaction temperature 60 °C, and reaction time 248.14 min. Under these conditions, the COD removal rate was 86.23%.

## 1. Introduction

In the past, as China’s steel industry continued to take off, it led to the development of the coke industry based on the coal chemical industry. For nearly 20 years, China’s coking industry has been leading the world. But China’s per capita water resources are very low, and the lack of water resources has obviously restricted the development of the chemical industry, especially for the current coal chemical industry [1,2,3,4,5]. 

There is a large amount of wastewater in the coking industry, and the organic matter in the wastewater is mainly phenols, which is difficult to degrade [6,7]. Inorganic matter in coking wastewater mainly refers to ammonia nitrogen content and sulfide. At the same time, its chemical oxygen demand (COD) and chroma are high, so it is an industrial wastewater that is difficult to treat [8,9]. The harm of coking wastewater mainly comes from phenols and ammonia nitrogen substances. Phenol-containing wastewater has great harm to the natural environment and the human body, both in the world and within China, it is the key control object. The higher the content of phenolic substances in water, the greater the harm, which will cause poisoning and death of organisms in the water. When phenols enter the human body, it will cause chronic poisoning and damage the central nervous system and liver function. Ammonia nitrogen substances will cause water eutrophication, resulting in ecosystem disorder and water quality deterioration, and making aquatic organisms unable to survive. The carcinogenic effect of ammonia nitrogen on the human body is mainly due to its instability, and it is easy to produce excessive carcinogens such as nitrosamines in water [10,11,12]. Table 1 is the water quality characteristics of typical coking wastewater.

Coking wastewater is generally treated by physical methods such as flocculation and sedimentation, and then by biochemical methods such as activated sludge or incineration [13]. Using the metabolism of anaerobic microorganisms and aerobic microorganisms, the method of decomposing organic pollutants in wastewater into CO_2_ and H_2_O is called biological treatment [14]. The activated sludge process is widely used in practical wastewater treatment, but it has the following problems. First, most coking plants in China are located in the northern region where winter is cold, and biological wastewater treatment requires higher environmental temperature. Then, the effective use of land is very important, and the biological method occupies a relatively large area. Finally, it is difficult to ensure the quality of coking wastewater treated by a single biological method, and the corresponding discharge standard cannot be reached [15,16]. In the 1950s, the incineration method was applied to wastewater treatment [17,18]. The principle of wastewater treatment by incineration is as follows. Wastewater is introduced into a high temperature incinerator to be completely gasified. Under the condition of introducing enough oxygen or air, the organic matter in the wastewater is burned into CO_2_ and H_2_O, and the inorganic matter is converted into ash [19]. The wastewater treated by this method has a very high COD removal rate. However, there are also the following problems in wastewater treatment by incineration. First, incomplete combustion of organic matter may generate harmful nitrogen oxides (such as NO, NO_2_, etc.), which will enter the air and lead to secondary pollution. Second, high temperature incinerators are difficult to popularize and apply on a large scale because of high material requirements and high energy consumption, which leads to high cost [20].

At present, many scholars are studying new processing technology [21,22], including electrochemical oxidation, supercritical water oxidation, catalytic wet oxidation, and photocatalytic oxidation. Among them, Catalytic wet peroxide oxidation (CWPO) is generally recognized. The method of catalytic wet oxidation is to put the reaction system under high temperature and high pressure and add an oxidant such as oxygen, air, or hydrogen peroxide [23]. Organic pollutants in wastewater, suspended oil, ammonia nitrogen, and other organic pollutants are oxidized into nontoxic and harmless small molecular substances [24] by catalytic wet oxidation. Under this technology, the pressure and temperature required for the reaction will be significantly reduced, and the reaction rate will become faster. At the same time, this technology has the advantages of fast oxidation rate, small floor space, environmental protection, and high treatment efficiency. CWPO [25,26,27] is a catalytic wet oxidation method. This method uses hydrogen peroxide as an oxidant, and the oxidant is mainly composed of hydrogen and oxygen. The main products after decomposition are H_2_O and O_2_ [28,29]. Compared with the traditional catalytic wet oxidation technology, the catalytic wet hydrogen peroxide oxidation technology has some advantages in industrial application and treatment results. Liquid hydrogen peroxide is the oxidant that catalyzes wet hydrogen peroxide oxidation technology. Compared with the ordinary oxidant air or oxygen, catalytic wet hydrogen peroxide oxidation can reduce the pressure of the reaction system [30,31]. This not only avoids the use of high pressure equipment and reduces the cost of equipment, but also allows the reaction to be carried out at normal temperature and pressure, thus reducing the loss and corrosion of equipment caused by high temperature and high pressure during the system reaction, and making the production process safe and low consumption. CWPO has a good effect on removing organic matter from wastewater and improving its biodegradability. This method has been widely accepted and studied in the reaction system of high-concentration organic wastewater.

Catalysts play an important role in CWPO organic wastewater treatment. Choosing the appropriate catalyst can not only reduce the activation energy of the reaction, but also improve the reaction rate and the effect of wastewater treatment. Because of the different phase states of the catalyst added in the system, the technology can be divided into homogeneous catalytic wet oxidation technology and heterogeneous catalytic wet oxidation technology. The catalyst used in homogeneous catalytic wet oxidation technology is a kind of biometal salt, which has solubility. The characteristic of this kind of biometal salt is that it can react with most organic substances in wastewater and decompose into small molecular inorganic substances, H_2_O, CO_2_, etc. [32,33]. However, it is difficult to recycle the catalyst in this method, which causes pollution, so this technology has not been widely used in industry. Heterogeneous catalytic wet oxidation technology is the most widely used technology to treat wastewater. The catalyst used is a solid catalyst, which can be recycled after the reaction is completed.

Montmorillonite (MMT) is a natural mineral of layered silicate, and its main mineral components are aluminosilicate and water, which has strong adsorption capacity and cation exchange performance. In the tetrahedron of montmorillonite, a part of Si^4+^ can be replaced by A1^3+^. In the octahedron, a part of A1^3+^ can be replaced by Zn^2+^ and Fe^3+^. Because some low priced positive ions replace high priced positive ions, the montmorillonite layers are negatively charged, and the excess negative charges can be compensated by cations Ca^2+^ and Na^+^ between the montmorillonite layers. In the formation of the montmorillonite granular layer, the compensation cations are distributed on the surface of montmorillonite and between layers, and are easy to exchange with inorganic cations, so that montmorillonite can introduce elements with catalytic activity to form pillared montmorillonite, and the spacing and surface area of montmorillonite layers can be expanded, thus making it a catalyst with high activity and stability.

Ai et al. [34] used CWPO to treat coking wastewater, the supported iron-activated carbon (Fe/AC) catalyst was obtained by the impregnation method. The COD of the treated wastewater was as high as 96.5%, and the catalyst had good stability. Sung et al [35] used copper pillared clay as a catalyst when treating reactive dyes, and then X-ray diffraction results showed that the specific surface area and interlayer spacing of the catalyst were large. Luo et al [36] used the ion exchange method to obtain Fe-Al-MMT as a catalyst when treating coking wastewater with CWPO, and the treatment result was a high COD removal rate. Aluminum-based composites have high specific surface area and high monolayer adsorption capacity, reaching 134.77 mg L^−1^ [37]. In recent years, the metals Mn, Co and Ce have been reported to have good catalytic synergy with Fe in different catalytic oxidation fields (such as electrocatalytic oxidation, ozone oxidation, photocatalytic oxidation, etc.) [38]. Damma et al. [39] prepared Fe/Ce/Co spinel catalyst by hydrothermal synthesis. CeO_2_ can promote the reaction involving the oxidation step, and shows high oxygen mobility when used as a carrier, which is helpful for improving the stability of the catalyst. 

In our preliminary research [40], a value of 5.5% total metal loadings of (Fe + Ce)/(Fe + Ce + Al) for Fe–Ce–Al/MMT catalysts has been determined. The present paper mainly investigates the effect of different Fe/Ce molar ratios on the performance of Fe–Ce–Al/MMT catalysts and their performance in treating actual wastewater.

## 2. Results and Discussion

### 2.1. Structural Analysis of Fe–Ce–Al/MMT Catalysts with Different Fe/Ce Molar Ratios

The structural analysis of Fe–Ce–Al/MMT catalysts with different Fe/Ce molar ratios is analyzed by N_2_ adsorption/desorption technology, and the corresponding N_2_ adsorption/desorption isotherms are shown in Figure 1. It can be seen that N_2_ adsorption/desorption isotherms of all Fe–Ce–Al/MMT catalyst samples belong to type IV as a whole, which shows that the Fe–Ce–Al/MMT catalyst after pillaring still maintains the mesoporous structure of raw montmorillonite. However, in the region where the relative pressure was lower than 0.45, the N_2_ adsorption/desorption isotherm of Fe–Ce–Al catalyst exceeds that of Type II, which indicates that the Fe–Ce–Al/MMT catalyst had some micropore structures, which further indicates that the raw material montmorillonite was successfully loaded. This just confirms the statement that montmorillonite will change from mesoporous structure to microporous structure in the process of pillaring, which was reported in the relevant literature. In addition, combined with Figure 1, it can be concluded that due to capillary condensation, when P/P_0_ is ≥0.45, the catalysts of Fe–Ce–Al-1/9 and Fe–Ce–Al-3/7 have H_3_ hysteresis loops, Fe–Ce–Al-5/5 and Fe–Ce–Al-7/3. The H_3_ hysteresis loop is usually caused by the pore structure of polymerized planar particles in the catalyst, while the H_4_ hysteresis loop generally belongs to the mesophase pore structure of the catalyst.

Figure 2 shows the pore size distribution curves of Fe–Ce–Al/MMT catalysts with different Fe/Ce molar ratios. As can be seen from Figure 2, the pore sizes of Fe–Ce–Al/MMT catalysts all show a unimodal distribution, and most of the pores are concentrated around 3.5 nm. Table 2 is the specific pore size distribution data. With the increase in Fe/Ce molar ratio, the pore size of the Fe–Ce–Al/MMT catalyst changes continuously. It can also be seen from Figure 2 that among all Fe–Ce–Al/MMT catalyst samples, the pore size distribution of the Fe–Ce–Al-7/3 catalyst is the widest, which indicates that the pore size of the Fe–Ce–Al-7/3 catalyst is larger than that of other catalysts. Combined with Table 2, it can be seen that 2–5 pores of Fe–Ce–Al/MMT catalyst account for a relatively large proportion. In addition, when the Fe/Ce molar ratio exceeds 7/3, the pore size of the Fe–Ce–Al/MMT catalyst gradually decreases with the Fe/Ce molar ratio. It just confirms the data analysis results of N_2_ adsorption/desorption isotherms. Therefore, the Fe–Ce–Al-7/3 catalyst has high catalytic activity because of its large specific surface area and wide pore size distribution.

### 2.2. Surface Morphology Analysis of Fe–Ce–Al/MMT Catalysts with Different Fe/Ce Molar Ratios

The surface morphology of Fe–Ce–Al/MMT catalysts with different Fe/Ce molar ratios was detected and analyzed by SEM, and the scanning patterns of all samples are shown in Figure 3. From Figure 3, it can be seen that the surface structures of all Fe–Ce–Al/MMT catalysts are basically similar, among which the surface structure of the Fe–Ce–Al-7/3 catalyst is the most regular, without any metal agglomeration, while the Fe–Ce–Al-9/1 catalyst obviously has some metal agglomeration, which may be because with the increase in Fe/Ce molar ratio, the Fe concentration in the column solution is too high, exceeding that of montmorillonite. This shows that the catalytic activity of the Fe–Ce–Al-7/3 catalyst may be the best among the five groups of catalysts.

### 2.3. Crystal Structure Analysis of Fe–Ce–Al/MMT Catalysts with Different Fe/Ce Molar Ratios

Figure 4 shows the XRD analysis of Fe–Ce–Al/MMT catalysts with different Fe/Ce molar ratios. It can be seen that the diffraction peak shapes of all catalyst samples are basically similar, which indicates that the crystal structure of montmorillonite has not changed much during the whole pillaring process. In addition, the characteristic diffraction peak of raw material montmorillonite appears around 2θ = 5. According to the diffraction angle, the d_001_ values of Fe–Ce–Al-1/9, Fe–Ce–Al-3/7, Fe–Ce–Al-5/5, Fe–Ce–Al-7/3, and Fe–Ce–Al-9/1 are 1.62 nm, 1.71 nm, 1.79 nm, 1.83 nm and 1.82 nm, respectively. This shows that the polycations of all catalysts successfully pillared into the layered structure of raw montmorillonite, thus increasing the interlayer spacing. At the same time, as can be seen from Figure 4, the diffraction peak of FeOOH appeared in the Fe–Ce–Al-7/3 and Fe–Ce–Al-9/1 catalyst at 2θ = 25, but no diffraction peak of cerium oxide appeared in the XRD patterns of all samples, which indicated that the pillared catalyst had higher metal dispersion than the catalyst prepared by the traditional impregnation method, and further indicated that the active metal content could not exceed that of Montmorillonite during ion exchange.

### 2.4. Analysis of Catalytic Performance of Fe–Ce–Al/MMT Catalysts with Different Fe/Ce Molar Ratios

Evaluation of catalytic performance and stability of Fe–Ce–Al/MMT catalysts with different Fe/Ce molar ratios was carried out under the conditions of initial pH = 4, H_2_O_2_ dosage of 13 mL/L, catalyst dosage of 3500 mg/L [34], reaction temperature of 60 °C, and reaction time of 30–240 min, with phenol removal rate and Fe ion dissolution as evaluation indexes. The specific evaluation results are shown in Figure 5.

As can be seen from Figure 5, with the increase in reaction time, the removal rate of phenol also increases. When the reaction time exceeds 3 h, the conversion rate basically tends to be stable, which shows that the optimal reaction time of the catalyst is 3 h, and the reaction is basically close to equilibrium after 3 h. When the molar ratio of Fe/Ce increases from 1/9 to 3/7, the removal rate of phenol also increases, and when the molar ratio of Fe/Ce continues to increase, the removal rate of phenol gradually decreases. The activity of the Fe–Ce–Al-7/3 catalyst is the highest, and the removal rate of phenol reaches 88.0% after 3 h of reaction. These results show that the high Fe/Ce molar ratio can make more Fe ions in the pillared solution column into the Keggin layered structure of montmorillonite. However, when the molar ratio of Fe/Ce is too large, too many Fe ions exceed the cation exchange capacity of the raw material montmorillonite, which will cause the remaining Fe ions to polymerize on the catalyst surface. Then these polymerized iron oxide clusters attached to the catalyst surface will be acidic and dissolved during the treatment of phenol by CWPO, and even cover the active sites of the catalyst, which will lead to the loss of some active components of the catalyst and a decline in catalytic activity, which is consistent with the above SEM and XRD characterization results. Figure 6 shows the effect of Fe–Ce–Al/MMT catalysts with different Fe/Ce molar ratios on Fe dissolution. It can be seen that with the increase in Fe/Ce molar ratio, the dissolution of Fe increases linearly, which is mainly due to the increase in Fe element in the catalyst with the increase in Fe/Ce molar ratio, which makes its relative content gradually increase in the reaction process.

In this section, the effects of total catalyst loading and Fe/Ce molar ratio on the structure and catalytic performance of Fe–Ce–Al/MMT catalyst were mainly investigated. The structure and catalytic performance of the Fe–Ce–Al-7/3 catalyst are better than other catalysts by XRD, BET, and SEM, and it is tested with the phenol solution.

### 2.5. Stability Experiment of Catalyst

Activity and stability are important indexes to evaluate the Fe–Ce–Al/MMT catalyst. In this experiment, the Fe–Ce–Al-7/3 catalyst (Fe/Ce molar ratio is 3/7, total loading is 5.5%, and calcination temperature is 500 °C) selected in the previous experiment was used to evaluate the catalyst stability. This section takes this catalyst as the research object, and continuously carries out phenol wastewater treatment five times. After 3 h of reaction, the stability of Fe–Ce–Al-7/3 catalyst was analyzed with Fe ion dissolution and phenol removal rate as evaluation indexes. The experimental results are shown in Table 3.

As can be seen from Table 3, after five reactions, the activity of the Fe–Ce–Al-7/3 catalyst basically did not decrease, the phenol removal rate remained basically unchanged, and the amount of Fe ions dissolved in a certain range did not change greatly, and with the increase in reaction times, the amount of Fe ions dissolved decreased, which indicated that some easily lost iron ions were dissolved first, and the later ones were more difficult to dissolve. From Table 3, it can be seen that the active components of the catalyst are less dissolved in each test, indicating that there is a strong interaction between the active components and the carrier montmorillonite, and the metal cations of the pillaring agent are exchanged with the cations between the montmorillonite layers, and the pillaring at a certain temperature also shows that the Fe–Ce–Al/MMT catalyst prepared in this study has good stability.

### 2.6. Quality Analysis of Coking Wastewater

Before studying coking wastewater treatment technology, the quality of coking wastewater should be analyzed first. That is, the composition, species, and content of organic matter in coking wastewater were studied. Table 4 lists the name and content of organic compounds in coking wastewater.

The water sample is extracted with dichloromethane solvent several times under neutral, alkaline, and acidic conditions, and the extracted samples are dehydrated and steamed, and analyzed with the GC/MS (QP2010 plus, Shimadzu Inc., Kyoto, Japan) Shimadzu GC-MS-QP2010plus. The test results show that the organic pollutants in the wastewater are mainly phenol and its derivatives, acids (benzoic acid and dodecanoic acid), aromatic hydrocarbons (naphthalene, benzene, toluene, xylene, and styrene), alkanes (eicosane), N-containing compounds (pyridine), and alcohols (octanol). After searching and comparing the relevant data in the library, 16 main compounds were detected, among which phenols and their derivatives were the main pollutants. As can be seen from Table 4, the content of phenol in the organic composition of coking wastewater accounts for 38.75%. The content of phenols and their derivatives accounted for 70.9%. These phenols and their derivatives mainly include phenol, methyl phenol, hydroquinone, ethyl phenol, and xylene.

### 2.7. Response Surface Experimental Design

Based on our preliminary research and the BBD center design principle, the COD removal rate was taken as the response value. Initial pH, H_2_O_2_ dosage, Fe^2+^ dosage, reaction temperature, and reaction time were taken as the influencing factors of the experiment. A total of forty-six groups of experiments with five factors and three levels were designed, including forty-six groups of factorial experiments and six groups of central point repeat experiments. The title, experimental design, and results of the response surface method (RSM) are shown in Table 5 and Table 6.

### 2.8. Analysis of Variance

After statistical analysis fitting by Design Expert, the quadratic multiple regression equation was obtained: COD removal rate = 85.70 − 0.17A − 0.21B + 0.56C + 0.38D − 0.11E + 0.015AC − 0.16AE + 2.500E − 003BD + 0.095BE + 0.010CD + 0.15DE − 0.17A^2^ − 1.34B^2^ − 0.83C^2^ − 5.625E − 003D^2^ − 0.43E^2^. Here, A denotes the initial pH, B denotes the amount of H_2_O_2_ (mL/L), C denotes the amount of Fe^2+^ added (mL/L), D denotes the reaction temperature (°C), and E denotes the reaction time (min).

At the same time, the analysis results of the Design Expert regression equation can be obtained, as shown in Table 7.

Through the analysis of variance in Table 7, it is concluded that the R^2^ of the model is 29.17/29.96 = 0.973632. It can be seen that the regression equation established by Design Expert has high fitting linearity and small experimental error, and the above model equation can be used to predict the experimental results. At the same time, the significance level of the model equation is less than 0.0001, which is highly significant. The significant levels of experimental influencing factors are as follows: the significance level of the initial pH value is 0.0009, and less than 0.05 means that the initial pH value is significant. The significance level of the addition of H_2_O_2_ is less than 0.0001, which indicates that the addition of H_2_O_2_ is highly significant. The significance level of the addition of Fe^2+^ is less than 0.0001, which indicates that the addition of Fe^2+^ is highly significant. The significance level of reaction temperature is <0.0001, indicating that the reaction temperature is highly significant. The significance level of reaction time is 0.0194, and less than 0.05 indicates that the reaction time is significant.

The F value reflects the contribution of experimental influencing factors to the model establishment. From the F value, it can be concluded that the influence of these five influencing factors on the phenol removal rate is in the order of Fe^2+^ addition > reaction temperature > H_2_O_2_ addition > initial pH value > reaction time.

### 2.9. Analysis of Response Surface Diagram

Box–Behnken response surface optimization analysis can obtain the 3D map of the response surface and the corresponding contour map. The 3D map and the corresponding contour map are used to intuitively analyze the interaction of various factors on phenol removal rate. The strength and magnitude of this interaction are reflected in the bending and steepness of the response surface. The larger the bending amplitude of the response surface formed between the two factors, the greater the interactive influence of the two factors on the phenol removal rate. On the contrary, it shows that the interaction between these two factors on phenol removal rate is not obvious. The relevant response surface diagram is as follows.

Through the intuitive analysis of the above Figure 7, Figure 8, Figure 9, Figure 10, Figure 11, Figure 12, Figure 13, Figure 14, Figure 15 and Figure 16, it can be concluded that the order of significance of the five factors affecting the COD removal rate of coking wastewater is the amount of Fe^2+^ > reaction temperature > the amount of H_2_O_2_ > initial pH value > reaction time. This is consistent with the conclusion of variance analysis.

The optimal experimental conditions of Box–Behnken response surface optimization analysis and the corresponding maximum COD removal rate are as follows: initial pH value is 3.46, H_2_O_2_ dosage is 19.02 mL/L, Fe^2+^ dosage is 5475.39 mL/L, reaction temperature is 60 °C, and reaction time is 248.14 min. The corresponding COD removal rate is 86.23%.

The experimental conditions of the verification experiment are as follows: the initial pH value is 3.5, the dosage of H_2_O_2_ is 20 mL/L, the addition amount of Fe^2+^ is 5500 mL/L, the reaction temperature is 60 °C, the reaction time is 250 min. Under these experimental conditions, we carried out five repeated experiments, and the results are shown in Figure 17.

As shown in Figure 17, under the optimized experimental conditions, the average COD removal rate of coking wastewater is 86.3%. It shows that the design of the response surface is scientific, and the optimal experimental conditions obtained by the response surface are reliable, and the purpose of prediction can be achieved by using it.

## 3. Experiment

### 3.1. Preparation of Fe–Ce–Al/MMT Catalysts

The actual wastewater comes from a coal plant in Yulin, northern Shaanxi Province. The main experimental materials and the detailed preparation process of Fe–Ce–Al/MMT catalysts have been described in detail in our preliminary research [34]. The obtained Fe–Ce–Al/MMT catalysts with different Fe/Ce molar ratios of 1/9, 3/7, 5/5, 7/3, and 9/1 are named Fe–Ce–Al-1/9, Fe–Ce–Al-3/7, Fe–Ce–Al-5/5, Fe–Ce–Al-7/3, and Fe–Ce–Al-9/1, in turn.

### 3.2. Characterization and Evaluation of Catalysts

In this experiment, different Fe/Ce molar ratios of Fe–Ce–Al/MMT catalysts were characterized by XRD, SEM, and N_2_ adsorption/desorption, and the performance for coking wastewater treatment was evaluated. Specific instrument types and evaluation experiment are shown in reference [34].

### 3.3. Detection of Phenol Removal Rate and Fe Ion Dissolution

GCMS-QP 2010 Plus gas chromatography–mass spectrometry produced by Shimadzu company in Tokyo, Japan GC/MS (QP2010 plus, Shimadzu Inc., Kyoto, Japan) was used to qualitatively and quantitatively analyze the removal rate of phenol from simulated phenol-containing wastewater. The ICP-OES (Optima, Perkin Elmer Instruments Co., Ltd., Waltham, MA, USA) were used to measure the ion content of the sample. The standard curves of phenol concentration and Fe ion dissolution were shown in our preliminary research [34].

### 3.4. Detection of COD

COD is very important for wastewater monitoring and is of great significance in wastewater detection indicators. It is the basis for measuring the amount of reducing pollutants in wastewater. In this experiment, the potassium dichromate method of GB 11914-1989 was used to determine the COD content in the actual coking wastewater [41].

The reaction principle of the potassium dichromate method is as follows:$${Cr}_{2}O_{7}^{2-}+6e+14H^{+}\to 2{Cr}^{3+}+7H_{2}O\;{Cr}_{2}O_{7}^{2-}+6{Fe}^{2+}+14H^{+}\to 2{Cr}^{3+}+6{Fe}^{3+}+7H_{2}O$$

The specific process of the potassium dichromate method is as follows. A 5 mL wastewater sample was put into a COD test tube, and 5 mL deionized water was added. Then, we added 0.2 g, 5 mL, and 15 mL of HgSO_4_, K_2_CrO_4_, and H_2_SO_4_ + AgSO_4_ solutions, respectively. We placed the COD detection tube on an energy-saving COD constant temperature heater, boiled it for 2 h, and naturally cooled it to room temperature. Then, we used 0.1 mol/L ferrous ammonium sulfate solution titration, and finally calculated the COD value of wastewater samples.

The calculation formula of the potassium dichromate method is as follows:COD (mg/L) = 8000C (V_1_ − V_2_)/V_0_
where V_1_ is the volume of ammonium ferrous sulfate standard titration solution consumed by the blank reagent, mL.

V_2_ is the volume of ammonium ferrous sulfate standard titration solution consumed by the sample, mL.

V_0_ is the volume of the sample, mL.

The value of 8000 is the conversion of the molar mass of 0.25 O_2_ in mg/L.

## 4. Conclusions

Based on the water quality analysis of coking wastewater and the experiment of simulating phenol wastewater, the response surface design experiment of coking wastewater was carried out. At the same time, compared with the results of phenol simulated wastewater, the conclusions are as follows:(1)We analyzed the components of coking coal wastewater, and the content of phenols and their derivatives accounted for 70.9% (mainly phenol). In addition, it also includes acids, aromatic hydrocarbons, alkanes, nitrogen-containing compounds, and alcohols.(2)Through the variance analysis of response surface, its 3D diagram, and the corresponding contour map, it is obtained that the order of five influencing factors on the COD removal rate of coking wastewater is as follows: the amount of Fe^2+^ > reaction temperature > the amount of H_2_O_2_ > initial pH value > reaction time. The optimum technological conditions of response surface were as follows: initial pH value was 3.46, H_2_O_2_ dosage was 19.02 mL/L, Fe^2+^ dosage was 5475.39 mL/L, reaction temperature was 60 °C, and reaction time was 248.14min. The corresponding COD removal rate is 86.23%.(3)The optimum process conditions verified by experiments are an initial pH value of 3.5, a dosage of H_2_O_2_ of 20 mL/L, an addition amount of Fe^2+^ of 5500 mL/L, a reaction temperature of 60 °C, and a reaction time of 250 min. Under these experimental conditions, the COD removal rate of simulated phenol wastewater finally reached 86.3%.

## Figures and Tables

**Figure 1 molecules-29-01948-f001:**
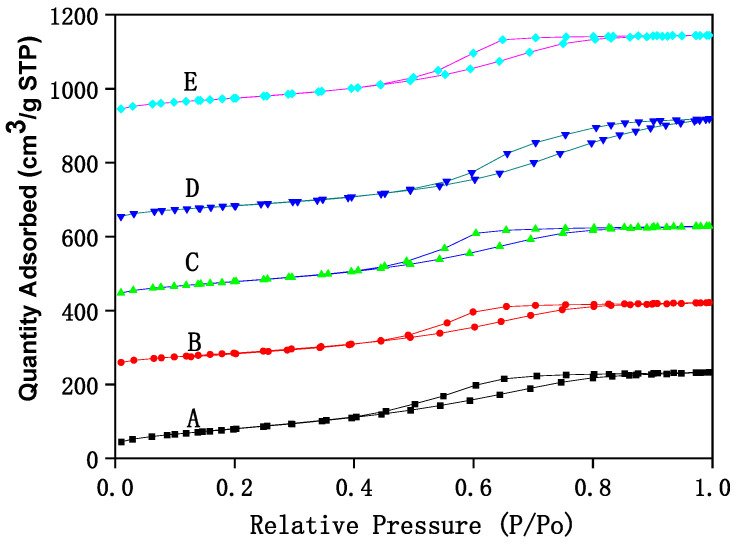
The N_2_ adsorption/desorption isotherms of Fe–Ce–Al/MMT catalysts with different Fe/Ce mole ratio: (A) Fe–Ce–Al-1/9; (B) Fe–Ce–Al-3/7; (C) Fe–Ce–Al-5/5; (D) Fe–Ce–Al-7/3; (E) Fe–Ce–Al-9/1.

**Figure 2 molecules-29-01948-f002:**
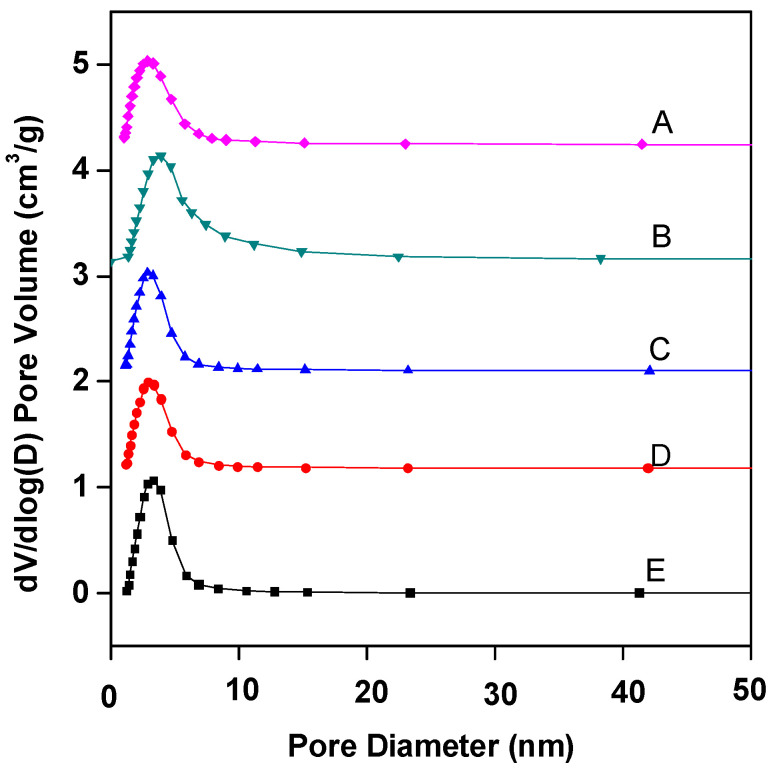
BJH adsorption pore distribution of Fe–Ce–Al/MMT catalysts with different Fe/Ce mole ratio: (A) Fe–Ce–Al-1/9; (B) Fe–Ce–Al-3/7; (C) Fe–Ce–Al-5/5; (D) Fe–Ce–Al-7/3; (E) Fe–Ce–Al-9/1.

**Figure 3 molecules-29-01948-f003:**
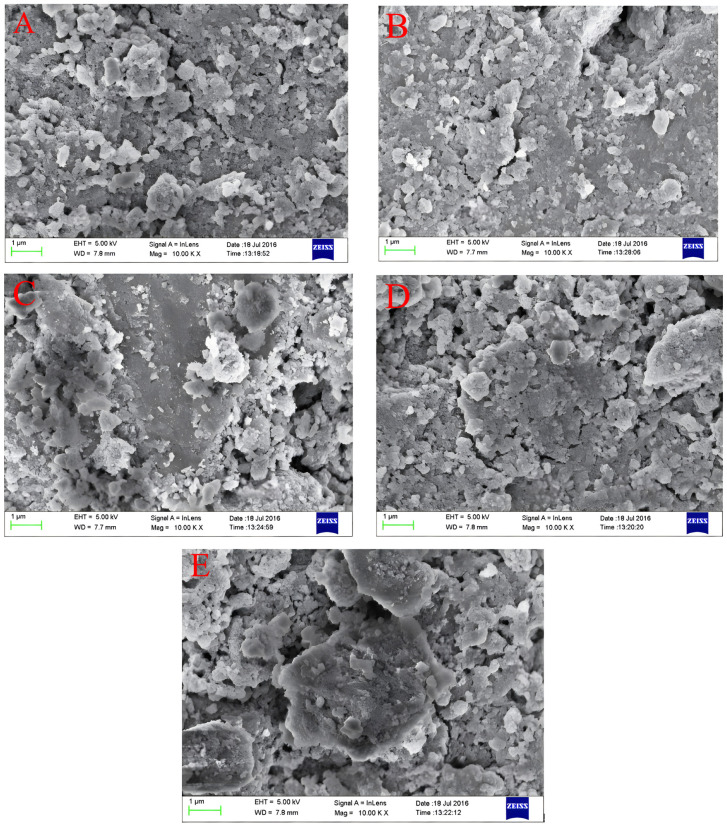
SEM microphotographs of Fe–Ce–Al/MMT catalysts with different Fe/Ce mole ratio: (**A**) Fe–Ce–Al-1/9; (**B**) Fe–Ce–Al-3/7; (**C**) Fe–Ce–Al-5/5; (**D**) Fe–Ce–Al-7/3; (**E**) Fe–Ce–Al-9/1.

**Figure 4 molecules-29-01948-f004:**
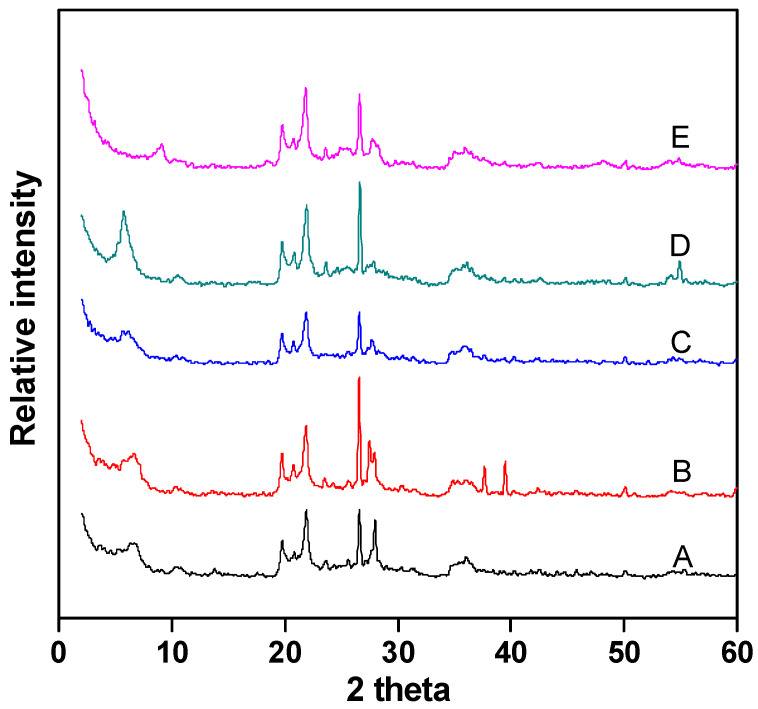
XRD patterns of Fe–Ce–Al/MMT catalysts with different Fe/Ce mole ratio: (A) Fe–Ce–Al-1/9; (B) Fe–Ce–Al-3/7; (C) Fe–Ce–Al-5/5; (D) Fe–Ce–Al-7/3; (E) Fe–Ce–Al-9/1.

**Figure 5 molecules-29-01948-f005:**
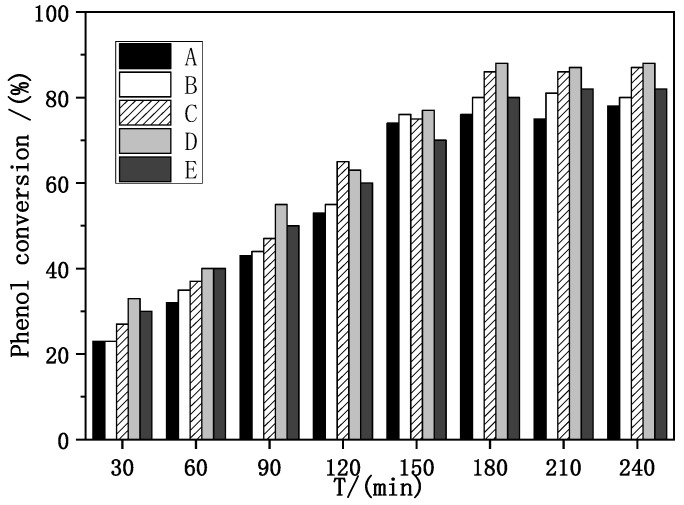
Effect of Fe–Ce–Al/MMT catalysts with different Fe/Ce mole ratio on phenol removal: (A) Fe–Ce–Al-1/9; (B) Fe–Ce–Al-3/7; (C) Fe–Ce–Al-5/5; (D) Fe–Ce–Al-7/3; (E) Fe–Ce–Al-9/1.

**Figure 6 molecules-29-01948-f006:**
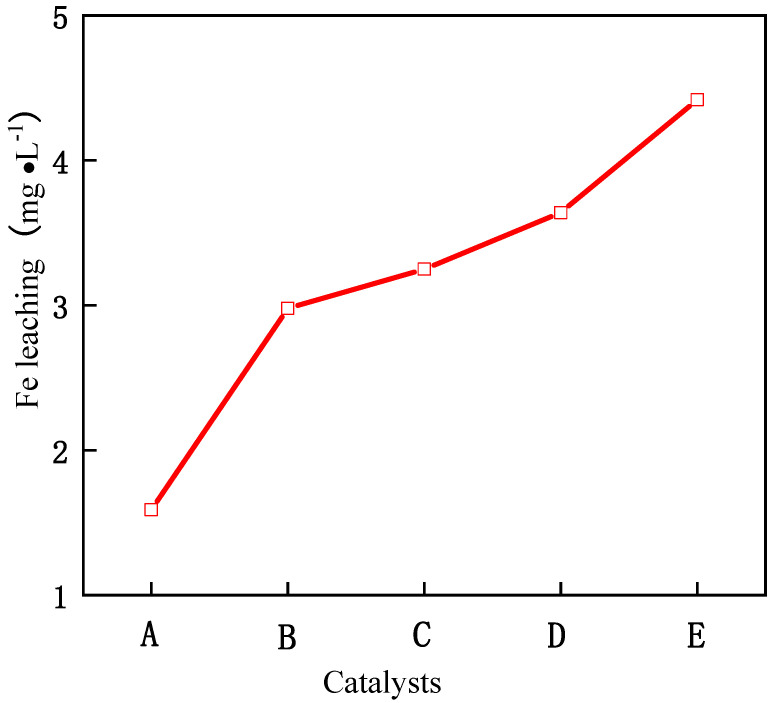
Effect of Fe–Ce–Al/MMT catalysts with different Fe/Ce mole ratio on Fe leaching: (A) Fe–Ce–Al-1/9; (B) Fe–Ce–Al-3/7; (C) Fe–Ce–Al-5/5; (D) Fe–Ce–Al-7/3; (E) Fe–Ce–Al-9/1.

**Figure 7 molecules-29-01948-f007:**
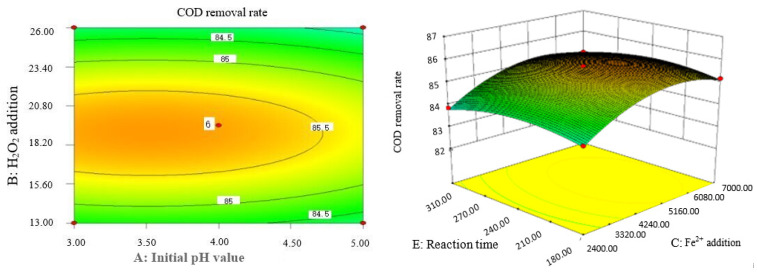
Influence of initial pH value and H_2_O_2_ dosage on the COD removal rate of coking wastewater.

**Figure 8 molecules-29-01948-f008:**
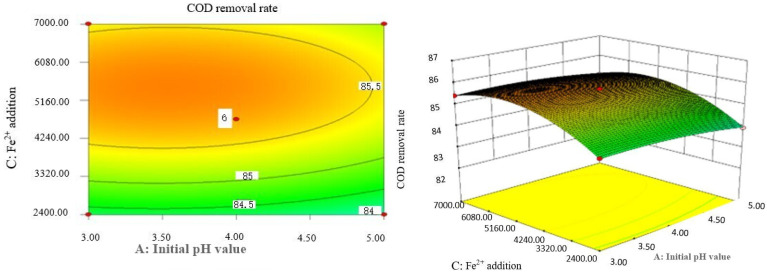
Effects of initial pH value and Fe^2+^ addition on COD removal rate of coking wastewater.

**Figure 9 molecules-29-01948-f009:**
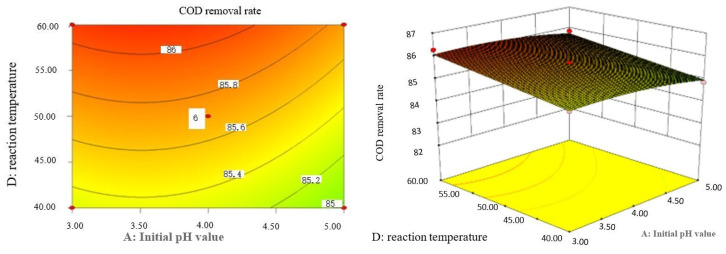
Influence of initial pH value and reaction temperature on COD removal rate of coking wastewater.

**Figure 10 molecules-29-01948-f010:**
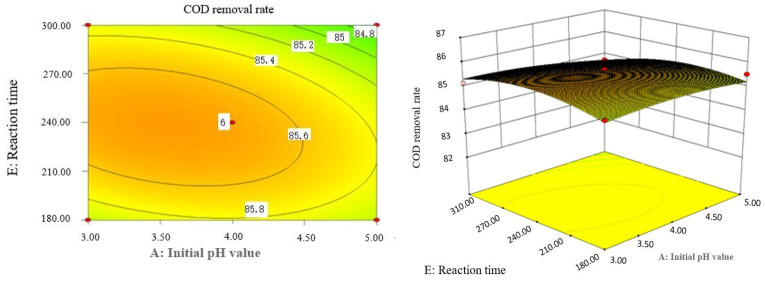
Effect of initial pH value and reaction time on COD removal rate of coking wastewater.

**Figure 11 molecules-29-01948-f011:**
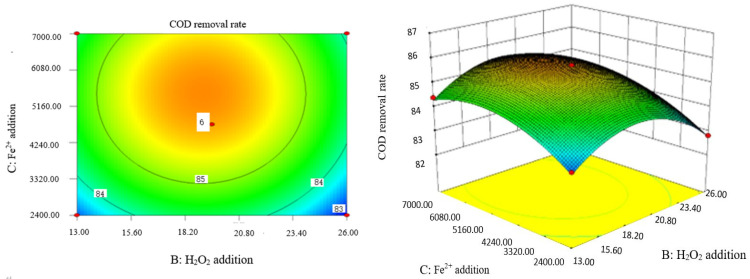
Influence of H_2_O_2_ and Fe^2+^ on the COD removal rate of coking wastewater.

**Figure 12 molecules-29-01948-f012:**
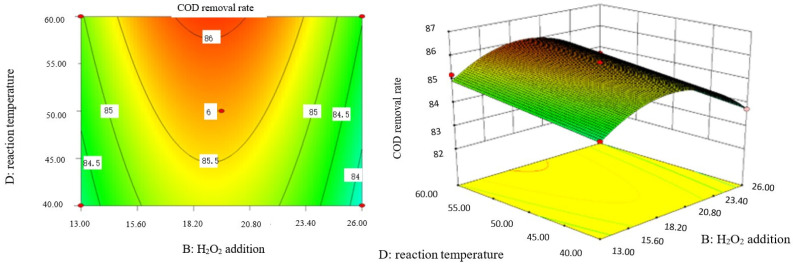
Influence of H_2_O_2_ dosage and reaction temperature on COD removal rate of coking wastewater.

**Figure 13 molecules-29-01948-f013:**
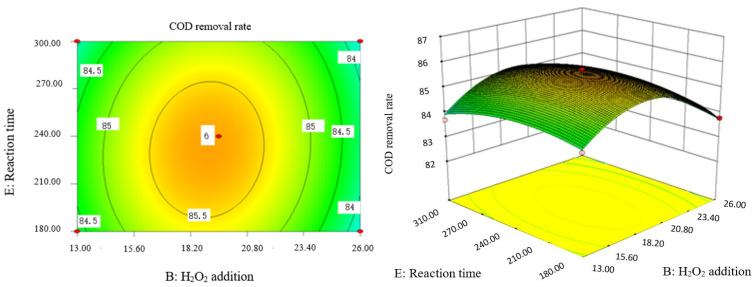
Influence of H_2_O_2_ dosage and reaction time on COD removal rate of coking wastewater.

**Figure 14 molecules-29-01948-f014:**
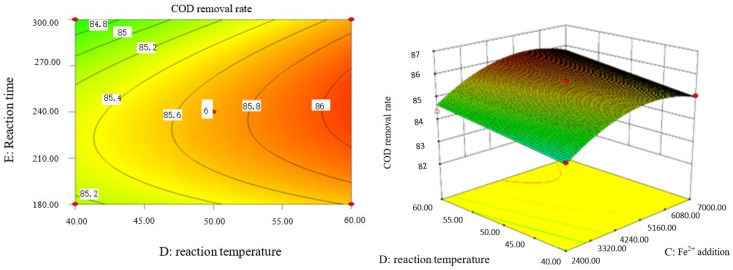
Influence of Fe^2+^ addition and reaction temperature on COD removal rate of coking wastewater.

**Figure 15 molecules-29-01948-f015:**
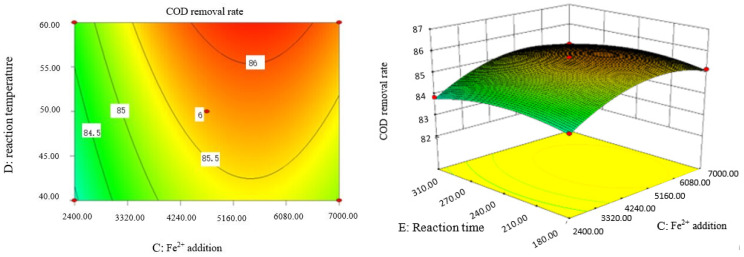
Influence of Fe^2+^ addition and reaction time on COD removal rate of coking wastewater.

**Figure 16 molecules-29-01948-f016:**
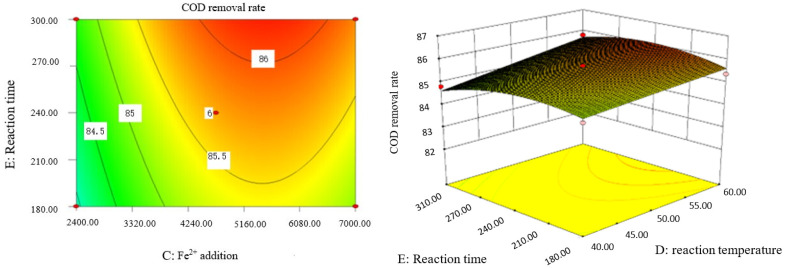
Influence of reaction temperature and reaction time on COD removal rate of coking wastewater.

**Figure 17 molecules-29-01948-f017:**
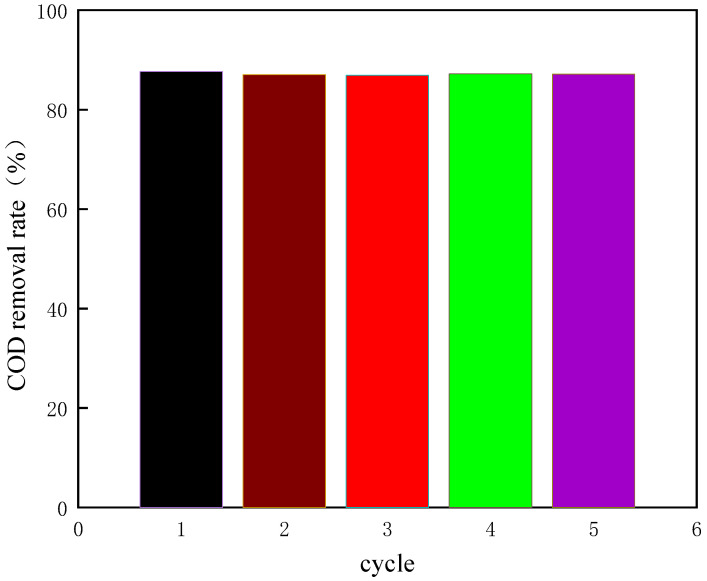
The COD removal rate of coking wastewater under optimal conditions.

**Table 1 molecules-29-01948-t001:** The water quality characteristics of typical coking wastewater.

COD (mg/L)	Ammonia Nitrogen (mg/L)	Volatile Phenol (mg/L)	pH
3434	549.3	483.0	8.2

**Table 2 molecules-29-01948-t002:** BJH adsorption pore distribution of Fe–Ce–Al/MMT catalysts with different Fe/Ce mole ratio.

Catalyst	BET /m^2^/g	Pore Volume /cm^3^/g	Average Pore Size /nm	Pore Distribution %		
				<2 nm	2–5 nm	5–20 nm
Fe–Ce–Al-1/9	153.7	0.1877	3.87	11.2	67.7	10.5
Fe–Ce–Al-3/7	167.1	0.1852	3.83	10.0	73.5	11.3
Fe–Ce–Al-5/5	178.3	0.1841	3.83	11.8	76.3	10.5
Fe–Ce–Al-7/3	180.9	0.1858	3.80	10.7	78.4	9.9
Fe–Ce–Al-1/9	165.5	0.1893	3.77	11.2	76.1	10.1

**Table 3 molecules-29-01948-t003:** Experimental results of stability evaluation of Fe–Ce–Al-7/3 catalyst.

Reaction Times	Fe Dissolution	Phenol Removal Efficiency
1	3.59	90.12
2	3.48	89.33
3	3.25	88.56
4	3.04	89.32
5	3.02	88.56

**Table 4 molecules-29-01948-t004:** The name and content of organic compounds in coking wastewater.

Peak Number	Name	Relative Amount	Retention Time	Absolute Intensity
1	toluene	0.911	4.716	185,187.917
2	xylene	0.723	7.473	146,954.537
3	styrene	0.148	8.134	30,199.756
4	benzene	0.104	10.425	21,135.765
5	pyridine	0.365	11.625	74,198.729
6	phenol	38.752	13.786	7,875,554
7	1-octanol	0.014	14.668	2747.649
8	2,5-xylenol	1.128	16.537	229,241.762
9	2,3-xylenol	1.972	17.186	400,766.626
10	naphthalene	0.086	17.66	17,569.104
11	Methylphenol	26.432	18.396	5,371,634.432
12	Ethylphenol	2.657	22.843	540,081.804
13	benzenediol	2.989	25.375	607,395.152
14	lauric acid	5.724	27.1256	1,163,259.684
15	1,2-phthalic acid	11.071	35.286	2,249,983.509

**Table 5 molecules-29-01948-t005:** Response surface analysis factors and levels.

Name	Unit	Form	Standard Deviation	Horizontal Downline	Horizontal Upper Line
Initial pH value		Factor	0	3	5
Dosage of H_2_O_2_	mL/L	Factor	0	13	26
Addition amount of Fe^2+^	mg/L	Factor	0	2400	7000
Reaction temperature	°C	Factor	0	40	80
Reaction time	min	Factor	0	180	300

**Table 6 molecules-29-01948-t006:** BBD design scheme and response values.

Serial Number	Initial pH	H_2_O_2_ (mL/L)	Fe^2+^ (mL/L)	T (°C)	Time (min)	Response Value (%)
1	4	19.5	4700	50	240	85.7
2	3	19.5	4700	40	240	85.28
3	4	26	7000	50	240	83.93
4	5	19.5	4700	50	300	84.72
5	4	26	4700	50	300	83.61
6	4	19.5	7000	60	240	85.51
7	4	13	4700	50	300	83.7
8	3	19.5	2400	50	240	84.34
9	4	19.5	4700	60	180	85.36
10	4	19.5	4700	50	240	85.71
11	5	19.5	7000	50	240	85.05
12	5	19.5	4700	50	180	85.51
13	3	19.5	4700	60	240	86.27
14	4	19.5	2400	60	240	84.39
15	4	13	4700	50	180	84.25
16	4	19.5	7000	50	300	84.99
17	4	13	7000	50	240	84.4
18	4	19.5	4700	50	240	85.7
19	4	19.5	4700	50	240	85.7
20	4	13	4700	60	240	85.2
21	5	19.5	2400	50	240	83.92
22	4	19.5	7000	50	180	85.17
23	3	19.5	4700	50	180	85.31
24	3	19.5	4700	50	300	85.14
25	5	19.5	4700	60	240	85.85
26	4	26	4700	40	240	83.74
27	3	26	4700	50	240	84.08
28	4	19.5	4700	40	300	84.81
29	4	19.5	4700	40	180	84.98
30	4	26	4700	50	180	83.78
31	4	19.5	2400	50	300	83.86
32	4	19.5	4700	50	240	85.7
33	4	26	2400	50	240	82.81
34	4	19.5	4700	60	300	85.79
35	4	19.5	2400	50	180	84.04
36	4	19.5	2400	40	240	84
37	3	13	4700	50	240	84.55
38	4	13	2400	50	240	83.28
39	5	13	4700	50	240	84.18
40	5	19.5	4700	40	240	84.86
41	5	26	4700	50	240	83.66
42	4	19.5	4700	50	240	85.7
43	4	26	4700	60	240	84.73
44	4	13	4700	40	240	84.22
45	3	19.5	7000	50	240	85.41
46	4	19.5	7000	40	240	85.08

**Table 7 molecules-29-01948-t007:** Analysis of the variance.

Source of Variance	Sum of Squares	Freedom	Mean	F	*p*
Model	29.17	20	1.46	45.95	<0.0001
A—Initial pH value	0.45	1	0.45	14.14	0.0009
B—H_2_O_2_	0.72	1	0.72	22.63	<0.0001
C—Fe^2+^	4.95	1	4.95	155.97	<0.0001
D—Temperature	2.35	1	2.35	73.99	<0.0001
E—Reaction time	0.2	1	0.2	6.24	0.0194
AB	0	1	0	0	1
AC	9.000 × 10^4^	1	9.000 × 10^4^	0.028	0.8676
AD	0	1	0	0	1
AE	0.096	1	0.096	3.03	0.0942
BC	0	1	0	0	1
BD	2.500 × 10^5^	1	2.500 ×10^5^	7.876 × 10^4^	0.9778
BE	0.036	1	0.036	1.14	0.2964
CD	4.00 × 10^4^	1	4.00 × 10^4^	0.013	0.9115
CE	0	1	0	0	1
DE	0.09	1	0.09	2.84	0.1047
A^2^	0.26	1	0.26	8.08	0.0088
B^2^	15.65	1	15.65	492.94	<0.0001
C^2^	6.01	1	6.01	189.32	<0.0001
D^2^	2.761 × 10^4^	1	2.761 × 10^4^	8.7 × 10^3^	0.9264
E^2^	1.59	1	1.59	50.01	<0.0001
Residual	0.79	25	0.032		
Misfitting error	0.79	20	0.04	2380.3	<0.0001
Pure error	8.3333 × 10^5^	5	1.667 × 10^5^		
Total	29.96	45			

## Data Availability

The original contribution put forward in the research is included in the article, and further inquiries can be directly contacted with the correspondent.

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
