# Peer review of "Study on the Performance Test of Fe–Ce–Al/MMT Catalysts with Different Fe/Ce Molar Ratios for Coking Wastewater Treatment"

_molecules, 2024, doi:10.3390/molecules29091948_

Round 1

Reviewer 1 Report

Comments and Suggestions for Authors

This work deals with the use of Fe-Ce-Al/MMT catalysts for the removal of coking wastewater by CWPO, which is innovative to some extent. Therefore, this manuscript can be published after considering the following problems.

1. This study should be a continuation of the author's published papers, and it should be noted that duplicated parts can be cited.

The author's published papers: Su, X., Wang, X., Li, L., Li, N., Liu, X., Zhang, P. Study on the performance test of Fe-Ce-Al/MMT catalysts for phenol-containing wastewater. AIP Advances, 2023, 13(10):105223.

2. It is suggested to revise and simplify the title of the paper.

3. The summary should be more concise and the format of references should be unified.

4 The introduction is a little fresh and wordy, which needs to be revised carefully.

5. In the conclusion section, the content needs refining.

6. The language of the full text is recommended for careful review and revision.

In lines 298, start the names of the sections by capital letters.

There are many inappropriate subscripts in the article, such as "P/P0" in line 261 and "N2- adsorption/desorption" in line 280. Please modify them along the article.

A lot of typographical mistakes have been found all along the text: the authors must make a deep revision of them and also of the English use in the manuscript.

In Figure 3, the author introduces the N2 adsorption/desorption isotherm in detail, but does not explain which material has the best effect in this test.

In line 322, why choose 3500mg/L catalyst?

In line 117, 2.5%, 4.0%, 5.5%, and 7.0% in weight or molar units?

In line 309, Please verify that FeOOH is spelled correctly.

In line 470 and 487, "Fig" should be replaced by "fig".

Comments on the Quality of English Language

This work deals with the use of Fe-Ce-Al/MMT catalysts for the removal of coking wastewater by CWPO, which is innovative to some extent. Therefore, this manuscript can be published after considering the following problems.

1. This study should be a continuation of the author's published papers, and it should be noted that duplicated parts can be cited.

The author's published papers: Su, X., Wang, X., Li, L., Li, N., Liu, X., Zhang, P. Study on the performance test of Fe-Ce-Al/MMT catalysts for phenol-containing wastewater. AIP Advances, 2023, 13(10):105223.

2. It is suggested to revise and simplify the title of the paper.

3. The summary should be more concise and the format of references should be unified.

4 The introduction is a little fresh and wordy, which needs to be revised carefully.

5. In the conclusion section, the content needs refining.

6. The language of the full text is recommended for careful review and revision.

In lines 298, start the names of the sections by capital letters.

There are many inappropriate subscripts in the article, such as "P/P0" in line 261 and "N2- adsorption/desorption" in line 280. Please modify them along the article.

A lot of typographical mistakes have been found all along the text: the authors must make a deep revision of them and also of the English use in the manuscript.

In Figure 3, the author introduces the N2 adsorption/desorption isotherm in detail, but does not explain which material has the best effect in this test.

In line 322, why choose 3500mg/L catalyst?

In line 117, 2.5%, 4.0%, 5.5%, and 7.0% in weight or molar units?

In line 309, Please verify that FeOOH is spelled correctly.

In line 470 and 487, "Fig" should be replaced by "fig".

Author Response

Thank you very much for your approval and careful comments on our manuscript. We have revised it carefully according to the revision opinions. Please refer to it.

Reviewer 2 Report

Comments and Suggestions for Authors

The authors synthesized the Fe-Ce-Al/MMT catalysts with different Fe/Ce molar ratios, characterized their partial physicochemical properties, and evaluated their catalytic activities for the removal of coking wastewater (phenol). This work contains some new results and could be considered for publication. However, the authors should revise their manuscript before acceptance for publication according to the following comments:

1.       What is the coking wastewater? And what is the CWPO? The full name of CWPO should be used in the title. Furthermore, the “removal rate” should be the “removal efficiency” or “phenol conversion”. The unit of a “rate” in catalysis should be “mol/s”, rather than “%”.

2.       The “PH” should be the “pH” in the abstract and main text. The suitable significant digits in all of the data (in all of the Tables) should be used, for example, 19.02 mL/L, 5475.39 mL/L, 86.2298% …

3.       Does the residual Mn species in the Fe-Ce-Al/MMT samples have any effect on catalytic activity of the samples?

4.       Are the Fe3+, Ce4+, and Al3+ ions incorporated into the lattice of calcium montmorillonite (MMT) or dispersed on the surface of MMT? What is the experimental evidence?

5.       The actual contents of Fe, Ce, and Al in the Fe-Ce-Al/MMT samples should be determined experimentally.

6.       What are the active sites or active species for the addressed reaction?

7.       What are the roles of Fe3+, Ce4+, and Al3+ in catalyzing the addressed reaction?

8.       What are the catalytic mechanisms?

9.       A comparison on catalytic activity of the as-obtained typical sample should be made with those of the related samples reported in the literature.    

Comments on the Quality of English Language

There are a large number of inappropriate English words or expressions in the manuscript. The authors should carefully polish the English of the whole manuscript.

Author Response

(The authors gave the same response as above.)

Round 2

Reviewer 2 Report

Comments and Suggestions for Authors

I have carefully checked the responses and modifications of the revised manuscript, and found that the authors have properly modified their manuscript according to the Reviewers' comments. Hence, I think that it is now acceptable for publication.